# Dynamic Target Pursuit by Multi-UAV Under Communication Coverage: ACO-MATD3 Approach

1st Zhuang Cao
*School of Information and Communication Engineering*
*Hainan University*
Haikou, Hainan
hnucz@hainanu.edu.cn

2nd Di Wu*
*School of Information and Communication Engineering*
*Hainan University*
Haikou, Hainan
hainuicaplab@hainanu.edu.cn

*Abstract*—This study proposes a new approach for cooperative pursuit of dynamic targets under communication coverage involving multi-unmanned aerial vehicles (UAVs). This approach combines the ant colony optimization algorithm with the multi-agent twin delay deep deterministic policy gradient, called ACO-MATD3. The ACO-MATD3 algorithm dynamically adjusts hyperparameters based on varying stages and requirements, greatly enhancing the stability and performance of cooperative multi-UAV pursuit tasks, especially under strong communication coverage. Experimental results demonstrate that the ACO-MATD3 algorithm significantly outperforms other algorithms in terms of mean reward and communication return.

*Index Terms*—Multi-UAV, Pursuit, Communication coverage, Ant colony optimization algorithm, Multi-agent reinforcement learning

## I. INTRODUCTION

In recent years, multi-unmanned aerial vehicles (UAVs) have found extensive applications in fields like agriculture [1], environmental monitoring [2], and communication [3], [4], due to their flexibility and ease of deployment. As technology progresses, UAVs are tasked with more complex challenges such as pursuing dynamic targets, where UAVs need to consistently pursue and approach a moving target in complex environments through strategic adjustments. This pursuit involves a strategic interaction between the UAVs and the targets, where effective decision-making is vital for success and showcases the UAV's intelligence. Therefore, developing effective pursuit strategies is crucial.

Significant research has been conducted on the pursuit of UAVs using traditional methods. For instance, the study in [5] developed a cooperative pursuit-evasion strategy for UAVs in a complex 3D environment, utilizing a heterogeneous system to enhance spatial perception and decision-making. However, this approach encounters challenges related to scalability, computational demands, and robustness in dynamic environments. In [6], the problem of minimizing the time for a UAV to pursuit a moving ground target by optimizing the pursuit strategy using sensor data. Additionally, a hierarchical game structure was

This work is partly distributed under the "South China Sea Rising Star" Education Platform Foundation of Hainan Province (JYNHXX2023-17G), the Natural Science Foundation of Hainan Province (624MS036), the Postgraduate Innovation Projects in Hainan Province (Qhys2023-290).

Corresponding author: Di Wu.

proposed in [7] to enhance the cooperative pursuit-evasion capabilities of UAVs in dynamic environments. Despite these advancements, the high computational complexity of these methods and the necessity to predefine the UAVs' flight paths limit their applicability in unknown environments.

Fortunately, advancements in deep reinforcement learning (DRL) have introduced new methods for addressing UAV pursuit problems. Techniques such as the deep deterministic policy gradient (DDPG) [8] and twin delay deep deterministic policy gradient (TD3) [9] enable simultaneous learning of value and policy functions, thereby enhancing algorithm efficiency and stability. However, in multi-agent environments, interactions between agents can lead to policy non-convergence when DRL algorithms are applied directly. To address this issue, multi-agent reinforcement learning (MARL) algorithms, including the multi-agent deep deterministic policy gradient (MADDPG) [10] and multi-agent twin delay deterministic policy gradient (MATD3) [11], have been developed. The MATD3 is an improvement based on MADDPG. These algorithms improve stability and collaboration among agents by employing a centralized training and decentralized execution (CTDE) mechanism [10].

Based on these DRL methods mentioned above, several studies have attempted to utilize DRL to solve UAV pursuit tasks. An approach proposed for UAV pursuit-evasion games utilizes hierarchical maneuvering decision-making with soft actor-critic algorithm [12] to enhance autonomy and strategic flexibility in complex environments. However, this method needs to work on high-dimensional state spaces. Another study [13] proposed a UAV pursuit policy combining DDPG with imitation learning to improve sample exploration efficiency, resulting in better performance and faster convergence than traditional DDPG method. A multi-UAV pursuit-evasion game was also explored in [14], utilizing online motion planning and DRL to enhance UAV interactions in complex environments. However, these studies still do not address the challenge of maintaining communication among UAVs while performing their tasks.

Based on the above related research, we propose an algorithm that combines MATD3 and ant colony optimization (ACO) algorithm to address the multi-UAV cooperative pursuit problem under communication coverage, called ACO-MATD3.

The algorithm can adaptively select the optimal hyperparameters at different stages during the training process. As a result, the multi-UAV system learns a policy that allows it to pursuit dynamic targets in the airspace without prior knowledge, while maintaining strong communication coverage from base stations (BSs). The main contributions of this paper are as follows:

(1) In contrast to non-learning based approaches [5], [6], [7], the problem of multi-UAV cooperative pursuit problem under communication coverage is formulated as a Markov game. Each UAV operates as an independent agent while cooperating with others to maximize cumulative rewards and optimize their policies.

(2) Differently from other DRL-based approaches [12], [13], [14], this study investigates the communication connectivity between multi-UAV and BSs during pursuit tasks, and considers the effect of noise in the environment on communication.

(3) Compared with the MATD3 [11] algorithm, the ACO-MATD3 algorithm proposed in this study can dynamically optimize the hyperparameters according to the training stage, reduce the impact of hyperparameters on performance, and improve training efficiency and effectiveness.

The paper is organized as follows: Section 2 provides the problem description and system modeling. Section 3 presents the ACO-MATD3 algorithm proposed in this paper. Section 4 analyses the results of the experiment. Section 5 concludes the paper.

## II. PROBLEM DESCRIPTION AND SYSTEM MODELING

In this section, we describe the multi-UAV pursuit problem under communication coverage. Then the BS antenna model and the path loss model are introduced. Finally, we illustrate the communication coverage model used in this experiment.

### A. Problem Description

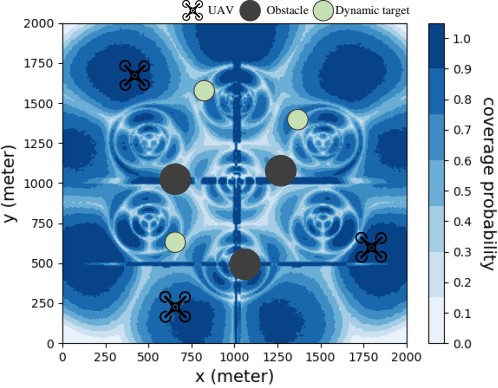

Fig. 1: Communication coverage strength map.

This experiment investigates the multi-UAV pursuit problem under communication coverage, consisting of multi-UAV, obstacles and dynamic targets, as shown in Fig. 1. Their initial positions are randomly generated. The BSs support UAV communication, with the blue shading in Fig. 1 indicating the strength of the communication coverage. During the pursuit

of dynamic targets, each UAV must avoid collisions with obstacles and maintain strong communication coverage.

### B. Antenna Model and Path Loss Model

This experiment formulates the antenna model of the BSs through the 3GPP [15] specification. Each BS has the same height $h_{BS}$ and divided into three sectors, each vertically placed with a uniform linear array of 8 elements.

The radiation pattern of each element is determined by combining its horizontal and vertical radiation patterns, defined as follows:

$$AH = -\min\left[12\left(\frac{\phi}{\phi_{3dB}}\right)^2, A_m\right] \quad (1)$$

$$AV = -\min\left[12\left(\frac{\theta - 90}{\theta_{3dB}}\right)^2, A_m\right] \quad (2)$$

where $\phi$ is the azimuth angle indicating the angle of the antenna in the horizontal plane, and $\theta$ is the elevation angle indicating the angle of the antenna in the vertical plane. Both are in degrees, $\phi_{3dB}$ and $\theta_{3dB}$ are the half-power beamwidths, $A_m$ is the element gain threshold.

The total gain of the antenna elements is expressed in dB as:

$$\begin{aligned} G_{ele_{dB}} &= G_{max} + A_{ele} \\ &= G_{max} + \{-\min[-(AH + AV), A_m]\} \end{aligned} \quad (3)$$

where $A_{ele}$ represents the power gain of the antenna element and $G_{max}$ is the maximum directional gain of the antenna element. For ease of computation, we convert $G_{ele_{dB}}$ to the linear scale of $G_{ele}$.

The combined gain of the antenna array is expressed in dB as:

$$G = 10 \times log_{10}|F_{ele} \times AF|^2 \quad (4)$$

where $F_{ele}$ is the arithmetic square root of $G_{ele}$, and $AF$ is the antenna array factor.

This experiment determines whether the communication link between the UAV and the BS sector is a line of sight (LoS) link or an non-line of sight (NLoS) link by assessing whether buildings in the environment obscure the communication link. The path loss of the LoS link from the UAV to sector $m$ is expressed in dB as:

$$h_m^{\text{LoS}}(t) = 28 + 22\log_{10} d_m(t) + 20\log_{10} f_c \quad (5)$$

where $d_m(t)$ represents the distance between the UAV and sector $m$, and $f_c$ denotes the carrier frequency.

The path loss of the NLoS link between the UAV and sector $m$ is given in dB as:

$$\begin{aligned} h_m^{\text{NLoS}}(t) = &-17.5 + (46 - 7\log_{10} h(t))\log_{10} d_m(t) \\ &+ 20\log_{10}(40\pi f_c/3) \end{aligned} \quad (6)$$

where $h(t)$ is the height of UAV at time $t$.

In addition, the channel small-scale fading is Rician fading in the case of LoS and Rayleigh fading in the case of NLoS.

## C. Communication Model

The baseband equivalent channel between the UAV and the communication BS sector $m$ at time $t$ is denoted by $H_m(t)$, where $1 \leq m \leq M$, and $M$ represents the total number of communication BS sectors linked with the UAV throughout its flight. The baseband equivalent channel $H_m(t)$ is influenced by the BS antenna array gain $G$, the path loss $\beta$, and the small-scale fading $h$. The magnitudes of $H_m(t)$ and $\beta$ are related to the position $q(t)$ of the UAV at time $t$, while $h$ is a random variable. The signal power received by the UAV from the communication BS sector $m$ at time $t$ can be expressed as:

$$P_m(t) = \bar{P}|H_m(t)|^2 = \bar{P}G_m(q(t))\beta(q(t))h(t) \qquad (7)$$

where $\bar{P}$ represents the transmit power of the BS sector $m$, which is assumed to remain constant. The path loss is calculated using the following equation:

$$\beta(q(t)) = \begin{cases} PL_{LoS}, & \text{if LoS link} \\ PL_{NLoS}, & \text{if NLoS link} \end{cases} \qquad (8)$$

where $PL_{LoS}$ and $PL_{NLoS}$ are the linear scales of $h_m^{\text{LoS}}(t)$ and $h_m^{\text{NLoS}}(t)$, respectively.

In this experiment, the signal to interference plus noise ratio (SINR) is used as a crucial criterion for evaluating the communication coverage performance of UAVs. This criterion can be expressed as:

$$SINR_t = \frac{P_m(t)}{\sum_{n \neq m} P_n(t) + \sigma^2} \qquad (9)$$

where $n$ represents the BSs not associated with the UAV at time $t$. In this case, the communication of the UAV is affected not only by interference from all non-associated BS sectors but also by the environmental noise, which impacts the quality of its communication.

To ensure communication coverage while the UAV is airborne, the SINR of the UAV should not drop below a minimum threshold $\alpha$. That is, the UAV is not under the communication coverage of the BS when $SINR(t) < \alpha$. Each UAV has an independent SINR at time $t$.

## III. MULTI-UAV COOPERATIVE PURSUIT USING ACO-MATD3

In this subsection, we characterize the UAV's state space, action space, and reward function within a Markov game framework and detail our proposed ACO-MATD3 algorithm.

### A. Markov Game with Multi-UAV

This subsection explores the framework of the Markov game as applied to multi-UAV systems. It details the state and action spaces for UAVs and defines the reward function guiding their interactions in a complex environment.

The state space for each UAV $i$ at time $t$ is defined as $s_{it} = (s_{ut}, s_{ot}, SINR_t)$, where $s_{ut} = (x_t, y_t, v_{xt}, v_{yt})$ is a combination of the position and the speed. Additionally, $s_{ot} = (l_{uu}, l_{uo}, l_{ut})$ represents the distance from the UAV to other

UAVs, obstacles and dynamic targets, respectively. $SINR_t$ denotes the SINR of the UAV at that moment.

The action space for each UAV is discrete. The action of UAV $i$ is defined as $V_u = (V_x, V_y)$, which denotes the vector velocity on the x-axis and y-axis, respectively. It also changes its own speed when the UAV collides.

The reward function for the UAVs in this experiment has three components. It encourages the UAV to quickly pursuit the dynamic target by considering the distance between them, providing a reward $R_{goal}$ upon successful pursuit. It also penalizes collisions to ensure safe flight and rewards higher SINR to promote flying in areas with better communication coverage. The reward function can be expressed as:

$$r(s_t, a_t) = R_{dist} + R_{goal} + R_{coll} + R_{SINR_t} \qquad (10)$$

### B. Fundamental of the ACO-MATD3 Approach

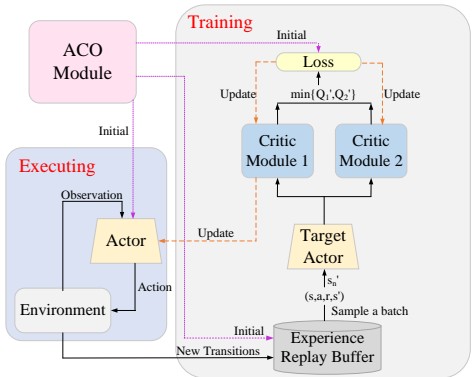

Fig. 2: Framework of ACO-MATD3 algorithm.

The ACO algorithm is an optimization algorithm that simulates the foraging behavior of ants. It directs the ant colony towards the optimal path in complex search spaces through pheromone accumulation and evaporation, combined with a probabilistic selection mechanism. This experiment combines the ACO algorithm with the MATD3 algorithm, aiming to dynamically choose the most appropriate learning rate $\alpha$, discount factor $\gamma$, and batch size $\mathcal{B}$ based on the current situation at different stages. This integration enhances the adaptability and robustness of the ACO-MATD3 algorithm. The framework of the algorithm is illustrated in Fig. 2.

We define a search space containing three hyperparameters: $\alpha$, $\gamma$, and $\mathcal{B}$. Each hyperparameter has multiple candidate values, and the range of values for these hyperparameters is given in detail in the next chapter. Additionally, we initialize a pheromone matrix.

In the initialization phase, we establish an initial colony of 100 ants. Each ant's hyperparameter configuration is derived by calculating selection probabilities based on the current values in the pheromone matrix. These probabilities then guide the random selection of hyperparameters from the correspond-

ing spaces. The selection probability for each hyperparameter value is calculated as follows:

$$p(v_i) = \frac{\tau(v_i)}{\sum_{k=1}^{n} \tau(v_k)} \tag{11}$$

where $p(v_i)$ represents the probability of selecting the $i$-th value, $\tau(v_i)$ denotes the pheromone level associated with the $i$-th value, and $n$ is the total number of possible values for the hyperparameter. This approach ensures that the search space is thoroughly explored, enabling the algorithm to evaluate a wide array of potential solutions right from the start.

In the multi-UAV system, each UAV uses hyperparameters derived from the current ant's configuration to execute a pursuit task, and the resulting reward values are recorded. If the reward from a particular set of hyperparameters exceeds the highest reward recorded in previous iterations, that configuration is designated as the optimal set for the current phase.

After each iteration, the pheromone level is adjusted according to the optimal hyperparameter configuration determined during the evaluation process. During this update process, the pheromone level for the chosen optimal configuration is increased to reinforce its selection in future iterations. Simultaneously, the pheromone levels for the other hyperparameters are reduced in accordance with the evaporation rate to ensure diversity in the search process and prevent premature convergence. This pheromone updating method can be succinctly described as follows:

$$\tau(v_i) \leftarrow \tau(v_i) + \Delta\tau \tag{12}$$

$$\tau(v_i) \leftarrow \tau(v_i) \times (1 - \rho) \tag{13}$$

where $\Delta\tau$ represents the increment added to the pheromone level upon a successful iteration, $\rho$ is the evaporation rate that moderates the decrease in pheromone levels to facilitate sustained exploration and exploitation balance. This dynamic adjustment ensures that the search algorithm not only intensifies exploration around proven successful parameters but also explores new potential areas effectively.

In the ACO-MATD3 algorithm, the target Q-value for UAV $i$ is calculated as:

$$y_i = r_i + \gamma \min_{j=1,2} Q_{w'_{i,j}}(x', a'_1, ..., a'_N) \tag{14}$$

where $r_i$ is the reward received by UAV $i$, $\gamma$ is the discount factor, $Q_{w'_{i,j}}$ is the $j$-th target critic network of UAV $i$, $x$ is the joint next state of all UAVs, and $a'_i$ represents the joint actions of all UAVs at the next time.

The loss function for updating the critic networks is:

$$L(w_i) = \mathbb{E}_{(x,a_i,r,x')\sim D}\left[(y_i - Q_{w_i}(x, a_1, ..., a_N))^2\right] \tag{15}$$

where $w_i$ represents the parameters of the critic network for UAV $i$, $D$ is the experience replay buffer.

The policy update rule for the actor networks is given by:

$$\nabla_{\theta_i} J(\theta_i) =$$
$$\mathbb{E}_{x,a_i\sim D}\left[\nabla_{\theta_i}\pi_{\theta_i}(s_i)\nabla_{a_i}Q_{w_i}(x, a_1, ..., a_N)\Big|_{a_i=\pi_{\theta_i}(s_i)}\right] \tag{16}$$

where $\theta_i$ represents the parameters of the actor network for UAV $i$, $s_i$ is the state of $i$th UAV, $\pi_{\theta_i}(s_i)$ is the policy of UAV $i$.

## IV. SIMULATION RESULTS AND DISCUSSION

### A. Parameter Setting

In this experiment, we build a 2 km × 2 km urban area scenario with numerous buildings, each with a maximum height $h_{bd}$ of 90 meters. The presence of a LoS link is determined by examining the linear connection between the BSs and the UAVs, considering the distribution of buildings. There are seven BSs in this area, totaling $M$ = 21 sectors. The transmit power of each sector is $\bar{P}$ = 20 dBm. The half-power beamwidth $\phi_{3dB}$ and $\theta_{3dB}$ both are 65°. The SINR interruption threshold is $\gamma_{th}$ = 1 dB. The noise power $\sigma^2$ of 5 dBm.

The hyperparameter search spaces for the ACO-MATD3 algorithm are: learning rate = $\{0.005, 0.01, 0.015\}$, discount factor = $\{0.93, 0.95, 0.97\}$, batch size = $\{512, 1024\}$. The remaining algorithm parameters and the parameters for the DRL algorithms are provided in Table I.

TABLE I: DRL algorithm parameters setting

| Definition | Value | Definition | Value |
|---|---|---|---|
| Max episodes | 100000 | Max step per episode | 25 |
| Replay buffer capacity | 1000000 | Batch size | 1024 |
| Learning rate | 0.01 | Gamma | 0.95 |
| R_coll | -2 | R_goal | 8 |

### B. Result Analysis

The experiment involves 3 UAVs, 3 dynamic targets, and 2 obstacles. To ensure fairness, all parameters were kept constant except for the ACO-MATD3 hyperparameter search space.

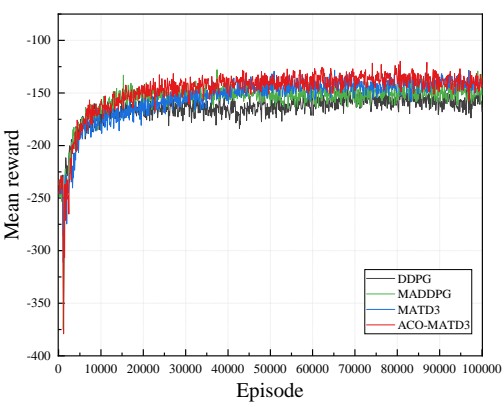

Fig. 3: Mean reward for different algorithms.

In Fig. 3, we compare the mean reward of the ACO-MATD3 algorithm with other algorithms. At the start of training, reward values drop significantly as the algorithms explore the environment to build awareness. It is clear from the figure that after reaching the converged state, the ACO-MATD3 algorithm achieves a higher mean reward than other algorithms. This

highlights the effectiveness of the ACO-MATD3 algorithm, which can dynamically select optimal hyperparameters at different stages, enhancing its performance in complex environments with communication coverage challenges.

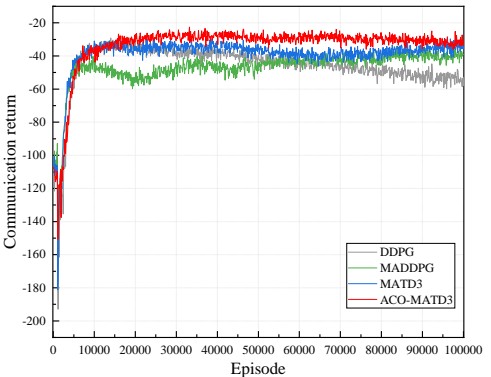

Fig. 4: Communication return for different algorithms.

The communication return for several algorithms are shown in Fig. 4. The final convergence values of the ACO-MATD3 algorithm are higher than those of the other algorithms, indicating that the flight path selected by the ACO-MATD3 algorithm for multi-UAV operations has stronger communication coverage. This further verifies the effectiveness of the ACO-MATD3 algorithm. In contrast, DDPG shows poor convergence performance in communication return because the UAVs operate independently and cannot learn a common policy. DDPG has poor convergence performance in communication return because the UAVs are all independent of each other and cannot learn a common policy. This situation highlights the improvement brought by the CTDE framework for multi-UAV cooperation.

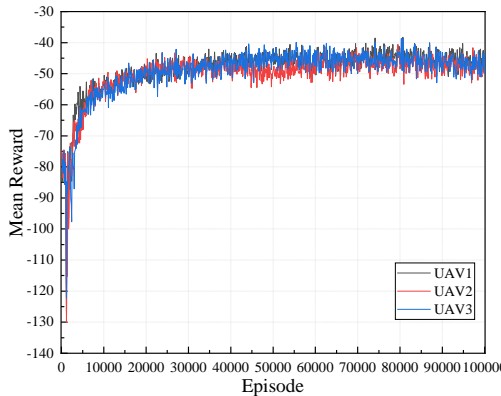

Fig. 5: Mean reward of each UAV in ACO-MATD3 algorithm.

Fig. 5 demonstrates the mean rewards of the three UAVs using the ACO-MATD3 algorithm in this environment. The convergence state aligns with the overall mean reward convergence of the ACO-MATD3 algorithm, demonstrating the superiority of this algorithm with the CTDE mechanism in coordinating the decisions of each UAV. This indicates that the ACO-MATD3 algorithm effectively optimizes both overall performance and individual UAV policies.

## V. CONCLUSION

In this study, we have presented the ACO-MATD3 algorithm to address multi-UAV pursuit of dynamic targets under communication coverage. This algorithm has dynamically adjusted hyperparameters for different stages to enhance performance and stability. Experimental results have shown that ACO-MATD3 outperforms other algorithms in mean reward and communication return, demonstrating the significant enhancement in task efficiency achieved through dynamically adjusting hyperparameters. Future research will explore how to safely conduct multi-UAV pursuit missions in more complex environments, especially those with dynamic obstacles.

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
