# OpenReview forum: "Dynamic Target Pursuit by Multi-UAV Under Communication Coverage: ACO-MATD3 Approach"
_IEEE.org/ICIST/2024/Conference — IEEE ICIST 2024 Conference Submission_

### Official Review · Reviewer_bxB4 · 2024-08-21
**This article is very interesting and a good one.**

**Rating:** 7
**Confidence:** 5

**Review:**

This study proposes a new approach for cooperative pursuit of dynamic targets under communication coverage involving multi-unmanned aerial vehicles (UAVs). The theory is correct and can be accepted after responding the following comments.
(1) In the introduction, it is not enough to state the current work. It should be expended and reconstructed.
(2) There are many typos and grammar errors. The authors should have a native English speaker or software packages to perform the editing check. For example, "To ensuring communication coverage..." should be revised to "To ensure communication coverage...".
(3) In the end of Section 1, the organization of this study is suggested to be summarized.
(4) The conclusion of the article suggests using the present perfect tense for description.

---

### Official Review · Reviewer_d5jf · 2024-08-21
**This article is quite fascinating and of high quality.**

**Rating:** 7
**Confidence:** 3

**Review:**

The paper titled "Dynamic Target Pursuit by Multi-UAV Under Communication Coverage: ACO-MATD3 Approach" proposes a novel cooperative pursuit of dynamic targets under communication coverage involving multi-unmanned aerial vehicles. It greatly improving the stability and performance of cooperative multi-UAV pursuit tasks, especially under strong communication coverage. My specific feedback is as follows: 1) In non-strong communication coverage, the stability and performance of cooperative multi-UAV pursuit task will be degraded or failed. Some discussion seems to be added. 2) Some formatting issues need to be addressed.

---

### Official Review · Reviewer_whuy · 2024-08-22
**this work is well organized and appears potentially interesting, it can be accepted with a little modification.**

**Rating:** 8
**Confidence:** 3

**Review:**

This study proposes a new approach for cooperative pursuit of dynamic targets under communication coverage involving multi-unmanned aerial vehicles. This approach combines the ant colony optimization algorithm with the multiagent twin delay deep deterministic policy gradient, called ACOMATD3. The ACO-MATD3 algorithm dynamically adjusts hyperparameters based on varying stages and requirements, greatly enhancing the stability and performance of cooperative multiUAV pursuit tasks, especially under strong communication coverage. In general, this work is well organized and appears potentially interesting, it can be accepted with a little modification.
1. What makes this system innovative compared to others?
2. Why was the ACO-MATD3 algorithm selected for this paper?
3. What are the future research directions discussed in the article?
4. Please analyze the simulation results in detail.

---

### Decision · Program_Chairs · 2024-09-06

Accept (Oral)